# Hunger and Health: Taking a Formative Approach to Build a Health Intervention Focused on Nutrition and Physical Activity Needs as Perceived by Stakeholders

**DOI:** 10.3390/nu13051584

**Published:** 2021-05-10

**Authors:** Kelsey Fortin, Susan Harvey

**Affiliations:** Department of Health, Sport and Exercise Sciences, School of Education and Human Sciences, Lawrence Campus, University of Kansas, Lawrence, KS 66045, USA; Suharvey@ku.edu

**Keywords:** food insecurity, hunger and health, nutrition, physical activity, health intervention, formative research

## Abstract

The intersections between hunger and health are beginning to gain traction. New interventions emphasize collaboration between the health and social service sectors. This study aimed to understand the nutrition and physical activity (PA) needs as perceived by food pantry stakeholders to inform a health intervention approach. The study used formative research incorporating mixed methods through surveying and semi-structured interviews with three food pantry stakeholder groups: Clients (*n* = 30), staff (*n* = 7), and volunteers (*n* = 10). Pantry client participants reported; high rates of both individual (60%, *n* = 18) and household (43%, *n* = 13) disease diagnosis; low consumption (0–1 servings) of fruits (67%, *n* = 20) and vegetables (47%, *n* = 14) per day; and low levels (0–120 min) of PA (67%, *n* = 20) per week. Interviews identified five final convergent major themes across all three stakeholder groups including food and PA barriers, nutrition and PA literacy, health status and lifestyle, current pantry operations and adjustments, and suggestions for health intervention programming. High rates of chronic disease combined with low health literacy among pantry clients demonstrate the need to address health behaviors. Further research piloting the design and implementation of a comprehensive health behavior intervention program in the food pantry setting is needed.

## 1. Introduction

Food pantries offer important resources in the federal aid system. Food insecurity is defined by the USDA as “limited or uncertain availability of nutritionally adequate and safe foods,” and 14 million U.S. households were food insecure in 2018 [1]. Despite the availability of federal nutrition assistance programs (e.g., SNAP, WIC, TANF), there is a gap in services, leaving organizations like Feeding America, a network of 60,000 food pantries and meal programs, still serving roughly 4.3 million meals to hungry people [2]. These emergency food services are reaching the most vulnerable populations needing both food and health services. The number of chronic diseases for adults in households with low food security, is on average, 18 percent higher than those with high food-security [2], and one out of three chronically ill food insecure adults are unable to afford medicine, food, or both [3].

Significant financial constraints leave food insecure individuals frequently limited to food pantry availability and low-cost food items. This translates into coping strategies promoting low nutrient diets high in processed foods [4]. In general, poor dietary intake (e.g., excess saturated or trans-fat intake, a diet low in fruits and vegetables) has been linked to a number of chronic diseases, including cardiovascular disease, Type 2 diabetes, some types of cancer and osteoporosis [5,6]. Overall, those living in food insecure households often have disrupted eating patterns and diets that are inadequate in nutrient-dense foods, contributing to malnourishment and an increased risk for poor health and chronic disease [7]. Beyond the quality of food, the existence of medical conditions associated with a poor diet can interfere with medication adherence [8]. Clients accessing a mobile food pantry reported that food insecurity impacts medication adherence due to the requirement that some medication be taken with food [9]. Therefore, gaps in the pantry schedule, lack of transportation or conflicting commitments may prevent individuals from accessing the food they need to meet medication recommendations.

Pantry clients have reported similar barriers, such as lack of transportation, inadequate kitchen equipment, lack of nutrition knowledge and skills, and few social support networks impacting their ability to eat healthy [10]. Other than adherence to medication for existing conditions, pantry clients are lacking in access, knowledge, and the to eat healthy diets to prevent disease onset. Begley et al. (2019) postulates that poor food and nutrition literacy behaviors contribute to food insecurity. Behaviors related to food planning and management, shopping, preparation, and cooking all show an association between food literacy behaviors and food security status [11]. In other words, the higher level of food literacy, the more food and nutritional behaviors individuals engage in that are associated with greater food security (e.g., food storage and preparation). Health literacy and self-efficacy have also been found to be predictors of food label use, which positively predict individuals diet quality [12]. As health professionals work to address hunger and health among food insecure populations, issues of food and health literacy are important interventional considerations.

The Department of Health and Human services (DHSS) recommends that American adults engage in a minimum of 150-minutes of moderately intense physical activity (PA) per week to experience health benefits [13]. PA rates among adults are low across the U.S. with nearly 80% of adults not meeting PA-recommended guidelines [13]. Common barriers associated with PA include a lack of confidence performing exercises, lack of time, lack of financial resources, and having diseases that create exercise limitations [13,14,15]. Food insecurity has demonstrated a significant association with adherence to PA guidelines among both adults and children [16]. Outside of the traditional barriers that prevent adults from engaging in PA, food insecure adults experience higher levels of stress and have poorer health, with a greater number of chronic diseases, creating larger obstacles to engaging in PA [16]. Additionally, food insecurity is associated perceptions, and readiness to engage in PA [17,18]. Within the context of disease, food insecure individuals report physical limitations that may prevent them from activities of daily life, including PA [19]. Connections between food insecurity and PA, particularly among adults, are the areas of hunger and health literature, which merit further research development.

According to the World Health Organization, non-communicable diseases (e.g., diabetes and cardiovascular disease) account for two-thirds of premature deaths worldwide [20]. Food insecure individuals are reporting broader disease prevalence and co-morbidities, such as obesity, disability and mental health disorders that warrant the need for a broader approach and multi-sector collaboration among medical providers, public health practitioners, social workers and food banks [21]. Within the space of chronic disease management, health coaching interventions have shown promise in the medical setting [22,23]. Health coaching often makes use of motivational interviewing techniques that promote collaboration, client evocation and autonomy, leading to successful behavior change across a variety of contexts, populations and health behaviors [24]. Health coaching uses a relationship building strategy in health behavior change through activities, such as health education sessions and individual practical support [25]. Health coaching shows positive results when targeting a range of diseases and populations, including diabetes, heart disease, hyperlipidemia and low-income patients [26]. If food pantries can implement a broader intervention design (e.g., health coaching), incorporating more holistic behavioral components (e.g., both nutrition and PA), can be developed that captures a broader range of food insecure individuals with comorbidities, and address lifestyle health behaviors leading to those disease.

The Academy of Nutrition and Dietetics released a position paper stressing the importance for nutrition practitioners to build partnerships with food pantries [27]. Health professionals are beginning to recognize the importance of targeted interventions among food insecure populations [28]. Recent intervention research relating to diabetes, nutrition education, and dietary and food purchasing behaviors within the food pantry setting resulted in positive health outcomes for food insecure pantry clients [29,30,31]. Among the literature, a recent systematic review of food pantry interventions revealed nutrition literacy and diabetic management interventions have been dominant in the field [32]. The cited studies indicate innovation and promise, yet present gaps in assisting individuals outside of the diabetic and nutrition scope. Only one study in the review of the literature utilized a more holistic health coaching approach within the food pantry setting [23]. Although, the study yielded positive pantry client health outcomes, it was still focused predominantly on nutritional behaviors, disregarding PA as an important disease prevention and management strategy.

As scientists and practitioners develop and implement interventions aimed at food pantry clients, little is known about the design and implementation of health intervention that combine both nutrition and PA health behaviors within a holistic health intervention model. This study uses formative research to understand nutrition and PA needs as perceived by food pantry stakeholders (pantry clients, volunteers, and staff) to inform a health intervention approach at a county-wide Midwest food pantry. This formative approach makes use of a community participatory model [33] to gain buy-in and consultation from the community of interest. The study aims to fill a gap in the literature by; (1) understanding more about PA behaviors and needs among food insecure adult pantry users; and (2) explore the program components of a comprehensive health intervention that incorporates both PA and nutrition as perceived by pantry stakeholders. This study will act as phase one to a multiphase intervention design research project.

## 2. Materials and Methods

The Institutional Review Board at a large Midwest research institution approved this study. Formative research using mixed methods incorporated surveys, individual interviews, and one focus group with three stakeholder groups (food pantry staff, volunteers, and clients). All data were collected on site at a local county-wide Midwest food pantry.

### 2.1. Pantry Context

The food pantry in the current study is the largest food pantry within the county it’s located serving roughly 13,000 residents in 2017 [34]. The county has a food insecurity rate of nearly 17% and overall poverty rate of 19% [34]. The pantry saw a 15% increase in overall client visits between 2017 and 2018 [35], with 51% of clients surveyed reporting having to skip meals between one and three times, on average, per week [31]. Demographically, over half of their clients (65%) identify as white and fall between the 18–64 age group (62%) [35]. Due to chronic disease concerns, with 62% of pantry clients surveyed reporting a household member with type 2 diabetes, the pantry has begun to offer health screenings on-site [35]. Additionally, the pantry offers a variety of nutrition programs including cooking and gardening classes, and an intensive culinary training program to encourage self-sufficiency among pantry clients [35]. The pantry utilizes seven full-time staff members and a fleet of volunteers.

### 2.2. Sample

Convenience sampling occurred focusing on three stakeholder groups (*n* = 47) (1) pantry staff (*n* = 7); (2) pantry volunteers (*n* = 10); and (3) pantry clients (*n* = 30). All staff currently employed by the pantry were included in the study, volunteers and pantry client participants were recruited until data saturation occurred. All participants were recruited in-person via direct communication with study staff during regularly scheduled pantry hours (M-F, 9 a.m.–5 p.m.). Inclusion criteria included individuals starting at age 18 to capture those of adult status and ending at age 75. This age range is representative of the majority age range of clients served (18–65) plus an extended age range (65–75) to capture the retired volunteer population. Additionally, stakeholder group classification (pantry staff, volunteer, or client), and ability to speak, read and write English were inclusion requirements. Participants incentives consisted of a “healthy eating goodie bag” containing a reusable grocery tote, cooking oil, one cooking utensil (wooden spoon, fork, or spatula), recipe cards, and informational brochures on various healthy eating topics. Only pantry clients were encouraged and received a “healthy eating goodie bag” upon completion of the study.

#### Demographic Characteristics

Table 1 displays pantry client (*n* = 30) demographics, and individual and household health status information. Majority of clients were Caucasian (80%, *n* = 24) and female (73%, *n* = 22). Disease prevalence was high with 60% (*n* = 18) reporting at least one chronic disease and 37% (*n* = 11) reporting more than one. Additional health status and demographic information is displayed in Tabe 1 below. Staff (*n* = 7) participant age ranged from 23 to 39, with majority of the participants (71%, *n* = 5) identifying as Caucasian/white, and two identified as mixed race. All staff work full-time, with years of experience ranging between one to six years. Lastly, Volunteer (*n* = 10) participants included individuals ages 18 to 79, primarily identifying as Caucasian (90%, *n* = 9), with one identifying as African American. Volunteer employment status ranged from full-time to retired.

### 2.3. Measures

Primary data collection involved three investigator-designed surveys and corresponding interview guides using a combination of newly developed questions based on the current study’s aims, and questions modified based on validated measures previously found in the literature. All survey measures were collected via hard copy, in-person, direct participant response. A researcher was present to answer participant questions.

#### 2.3.1. Client Survey Measures

The client survey included validated measures through questions on self-reported health [36] and the Behavioral Risk Factor Surveillance Survey nutrition and PA module measures [37]. Investigator-designed measures included categorical questions (yes or no) on individual and household chronic disease diagnosis (e.g., diabetes), and barriers to healthy eating (e.g., healthy foods are too expensive) and PA (e.g., I don’t know enough about physical activity). Last, the survey asked participants to report individual demographic characteristics (race/ethnicity, gender, annual household income, employment status, and level of education). The survey consisted of 28 questions and the full details can be reviewed under Appendix A.

#### 2.3.2. Client Semi-Structured Interviews

An investigator-designed moderator’s guide, which corresponded with survey questions, guided semi-structured interviews. Sample questions included, “What are some of the challenges and barriers to choosing and cooking healthy options?” and “What current health issues are you and/or members of your household facing? Last, questions pertaining to intervention components included “What do you think are some critical characteristics of this program? (Probe: How often meetings are, time of day, days of the week, how long, educator characteristics, location, electronic vs. in-person)?” The moderator’s guide consisted of 14 questions and the full details can be reviewed under Appendix A.

#### 2.3.3. Volunteer/Staff Survey Measures

Volunteer and staff measures included an investigator-designed survey informed by the study aims and topics represented in the client survey. Example questions include categorical questions (often, sometimes, never) related to client engagement within the topics of health, nutrition and PA (e.g., “How often do you engage with clients about the cost of food?). Last, the survey asked participants to report individual demographic characteristics (race/ethnicity, gender, annual household income, employment status, level of education and number of years of service at the current food pantry). The survey consisted of 16 questions and the full details can be reviewed under Appendix A.

#### 2.3.4. Volunteer/Staff Semi-Structured Interviews

Interviews consisted of an investigator-designed moderator’s guide corresponding to the survey. Sample questions included, “What questions do clients most commonly ask about (a) Food/food products, (b) Nutrition, (c) Physical Activity, (d) Health (e) Programs/resources offered by the pantry” and “What important topics within nutrition, physical activity, and health should be covered in an intervention program?” The moderator’s guide consisted of 8 questions and the full details can be reviewed under Appendix A.

### 2.4. Data Collection

Participants completed a written informed consent prior to data collection. Collection occurred through in-person hard copy surveys completed by participants, individual semi-structured interviews with pantry clients and volunteers, and one focus group with pantry staff.

#### 2.4.1. Participant Surveys

Survey responses were collected from all three-stakeholder groups (staff, volunteers, and clients) immediately before conducting interview questions. Surveys were administered in hard copy using paper and pencil, and were completed independently by study participants. Study staff were available for participant support. 

#### 2.4.2. Participant Interviews and Focus Group

Volunteer and client groups participated in follow-up individual semi-structured interviews, while staff participated in a single focus group during a routine staff meeting. All interviews and focus groups were semi-structured, immediately followed survey completion, and were located in a secure private room on-site at the food pantry. All correspondence was audio recorded with sessions lasting between roughly 30 to 60 min in length. A single investigator (the PI) with training and experience in qualitative methods and the interview protocol conducted interviews and took field notes. Member checks and debriefings occurred during interviews to ensure accuracy of participant statements and to increase trustworthiness [38]. 

### 2.5. Data Analysis

All data were reviewed and analyzed separately, then brought back together to find convergent themes across all sources and stakeholder groups. All survey responses were input into IBM SPSS Statistics 26 software for descriptive data analysis.

#### Interview/Focus group Analysis

All interviews were audio recorded and transcribed verbatim by the PI of the study. Once transcribed, a priori categories, based on categories within the semi-structured interview guide, directed the initial coding process and were combined with exploratory findings to generate final themes [39]. Last, data triangulation occured between the existing literature, stakeholder surveys and stakeholder interviews/focus group to informed research findings [39,40]. This process included two co-investigators of the research team.

## 3. Results

This section will provide a detailed description of each stakeholder group’s results separately, followed by a joining of the data generating final convergent themes. Final convergent major themes include food and PA barriers, nutrition and PA literacy, health status and lifestyle, current pantry operations and adjustments and suggestions for health intervention programming.

### 3.1. Client Results

Client survey responses revealed low consumption of fruits and vegetables with over half (67%, *n* = 20) reporting zero to one servings of fruits per day, and 47% (*n* = 14) reporting zero to one servings of vegetables per day. Commonly reported healthy eating barriers include: healthy food being too expensive (40%, *n* = 12), not knowing enough about healthy cooking (37%, *n* = 11), not knowing enough about general nutrition to make healthy meals (30%, *n* = 9), and not knowing how to choose and store fresh produce (27%, *n* = 8). A high rate of participants (67%, *n* = 20) reported low PA between zero to 120 min per week. Common barriers preventing participants from engaging in regular PA, included having health conditions that restrict activity (30, *n* = 9), lack of enjoyment for PA (27%, *n* = 8), lack of access to a facility to engage in PA (23%, *n* = 7), and having a job that is physically demanding (20%, *n* = 6).

During client interviews, four themes emerged, including Food and PA barriers, Nutrition and PA literacy, Health Status and Lifestyle, and Suggestions for Health Intervention Programming. In the first major theme, participants reported things such as cost, and food preparation restrictions as leading roadblocks to improving nutrition. One participant reported, “Right now, I live in a camper out in the park, and I don’t have electricity in it, so mostly it’s the food banks, or going to get something that’s cooked in the store. Unless I can get a fire going, so that limits me and what I can do.” Barriers to PA, included mental and physical limitations and occupational restrictions. Occupational restrictions include sedentary jobs and lack of time due to multiple jobs.

The second major theme, Nutrition and PA Literacy is associated with general nutrition and PA education. Within nutrition, categories, such as cooking, specialty diets, and produce storage and preparation were noted. Within PA most feedback was focused on general strength exercises, and exercises for physical limitations. Participants also mentioned a desire for weight management education with statements like, “Losing weight. What I would need to do to really lose some weight. And not just do strange starving eating type of things. The healthy way to do it.”

Third, the Health Status and Lifestyle theme corresponded with the depth of physical and mental illness across participants with one participant sharing “I have anxiety, depression, migraines, frontal lobe seizures, turrets, treated for blood clots, get treated for low vitamin B, Arthritis.” Additionally, there were reports on impacts to lifestyle due to disease. These related to both positive impacts, such as disease translating to improvements in health behaviors, and negative impacts with connections to disease affecting quality of life in examples like “She took me off work for two months to see if we could get it under control [high blood pressure], so hopefully.”

In the fourth major theme, Suggestions for Health Intervention Programming, pertained to programmatic and structural recommendations from pantry clients. Structurally, participants were interested in both electronic services and face-to-face services, as well as group and individual formats. They reflected on the idea of social support from both the health educator or coach, and other pantry client intervention participants in a group setting. Recommendations for intervention content included statements such as “It should be the holistic approach. Teaching people to eat better sooner, instead of waiting until the point of diabetes or the health issues.” Last, participants demonstrated support and excitement for a health intervention program by stating comments, such as “I think this is a fabulous idea I think it is doable with a lot of your hard work and I look forward to you moving forward and changes ahead.”

### 3.2. Staff Results

Staff survey responses demonstrated that within the category of health, staff reported often engaging with clients about health insurance (29%, *n* = 2) and local health services (29%, *n* = 2). Within nutrition, staff reported often engaging with clients about cost of food (86%, *n* = 6), quick meal options (57%, *n* = 4), and food restrictions (57%, *n* = 4). Within PA, staff indicated often engaging with clients about physical limitations (57%, *n* = 4).

Staff participated in a follow-up focus group instead of interviews to capture collaborative staff ideas as a part of a monthly staff meeting. Three themes emerged. Themes included Specialty Diet Questions, Pantry Operations, and Client Education. Specialty Diet questions included clients coming in with specific recommendations from medical providers with one staff member reporting, “I am finding more hyper specificity. [clients reporting] This is my diet, I have talked to my doctor, and they say I need to be eating these specific items, do you have any of those?”

Within the theme of Pantry Operations, staff proposed a variety of pantry operational changes that may assist clients with questions and food choice. This theme included creating general handouts, nutritional nudges, and increased meal kit options. Last, the Client Education theme, informed by direct client experience and observations, led to recommending general nutrition and PA education. For example, one staff member said the following: “Helping people understand how to be more realistic [portion size], my immediate thought goes to My Healthy Plate campaign.”

### 3.3. Volunteer Results

Volunteer survey responses demonstrated that within the category of health, volunteers reported often communicating with clients about high blood pressure (30%, *n* = 3) and local health services (30%, *n* = 3). Volunteers reported sometimes engaging about unusual food items (50%, *n* = 5), building healthy meals (*n* = 4), and food storage (*n* = 4). Volunteers indicated never engaging with clients about PA in nearly all categories.

During semi-structured interviews, four themes emerged including Pantry Questions, Pantry Shopping Adjustments, Client Education, and Volunteer Training. Within the theme of Pantry Questions, volunteers highlighted frequent client questions related to either food products or preparation. One volunteer indicated: “Sometimes people will ask about what would be a good way to prepare this vegetable or meat,” or pantry logistics “not too many questions other than how many points is this [food item]”.

Volunteers offered recommendations for Pantry Shopping Adjustments addressing the topics of food products/preparation and pantry logistics. Recommendations included adding information for use and preparation of unusual produce and including simple recipes directly with these items. Major topics highlighted within the Client Education theme included general nutrition and PA guidelines with comments like, “Most people don’t have a general understanding of nutrition,” and shopping strategies, “Educating on how to effectively use their points. Some people only have 10 points and they get 4 sandwiches and that is going to last you a max of 2 days.”

The Volunteer Training theme emphasized conflicting opinions. Regarding volunteer training, some volunteers indicated interest in receiving training related to “Food stamp options. How or where; opportunities to talk about options for food,” with other volunteers indicated a lack of interest in further training with rationales like, “A lot of us are retired and not wanting to fill that role [health specific volunteer role].”

### 3.4. Final Convergent Major Themes

There were five identified final convergent major themes including Food and PA Barriers, Nutrition and PA Literacy, Health Status and lifestyle, Current Pantry Operations and Adjustments, and Suggestions for Health Intervention Programming.

Food and PA Barriers, include identification of life circumstance that make healthy eating and PA difficult among pantry clients. Barriers that were reported included cost of food, produce storage and self-life, physical limitations to exercise, and the perception that PA is a privilege based on social status.

Nutrition and PA Literacy, the second theme, pertains to gaps in knowledge about healthy eating, selection and preparation of foods, PA recommendations based on limitations both identified by the clients through personal experience, and volunteers and staff based on client interactions and questions.

Similarly, clients’ personal reports, and volunteer and staff interactions with clients demonstrate how food insecurity and limitations due to disease influence clients lives under the Health Status and Lifestyle major theme. This included reporting on how much disease clients were experiencing daily and coping strategies such as seeking out dietary recommendations from staff at the pantry.

The fourth major theme, Current Pantry Operations and Adjustments, relates to volunteer and staff experience with the current climate within the pantry associated with nutrition and PA among clients, and ideas for adjustments to create a more informed and positive experience. This included ideas for inclusion of nutrient information in meal kits and throughout the pantry, as well as guidance on how to use their pantry points and potential training opportunities for volunteers.

Suggestions for Health Intervention Programming highlights the perspectives from all three stakeholder groups related to intervention program components consisting of nonjudgmental, supportive, coaching, with the inclusion of PA and nutrition education, and support for hosting such a health intervention program in the pantry setting. A summary of these final convergent themes and corresponding client, volunteer, and staff quotes can be found in Table 2.

## 4. Discussion

Consistent with previous research, pantry clients reported high levels of individual and household chronic diseases [40,41], which are compounded by client reported gaps in doctors’ visits and health insurance coverage [41]. As more research connects the dots between food insecurity and insufficient medical care, organizations work to provide solutions in both pantry and clinical settings. Within clinical settings, screenings, referrals, and connecting patients with emergency food services is becoming a more common practice [19]. Additionally, interventions in the form of food pharmacy programs are connecting patients with food and nutrition resources within medical facilities [42,43]. Medical interventions are surfacing and have shown promise in food pantry settings [44]. Within the pantry setting, disease specific interventions (e.g., diabetes management interventions) have shown success among pantry clients [22,45]. However, disease specific interventions leave an unmet gap in serving pantry clinics with co-morbidities outside of the scope of that intervention. Additionally, little is known about targeting nutrition and PA behaviors in a holistic health intervention framework to address chronic disease among food pantry users. Health coaching frameworks with the use of motivational interviewing techniques have demonstrated effectiveness in chronic illness management [46]. Only one study was found with the employment of health coaching as a component of a more comprehensive intervention model within the food pantry [28]. By providing interventions around a health coaching framework, using a combination of health education and motivational interviewing, health coaches can address a broader range of clients providing clients with both nutrition and physical activity education, and social support, thereby increasing self-efficacy [47].

All three-stakeholder groups identified poor nutrition and PA literacy as a contributor to poor health outcomes. Research has shown low food and nutrition literacy may contribute to food insecurity in developing countries [15], while health literacy and self-efficacy have been found to predict food label use, which is positively related with diet quality [16]. As health education contributes to relationship building between health coaches and patients [30], further education through health intervention programming using these program components within the pantry setting could lead to improvements in food security status and diet quality [15,16]. The lack of skills in preparing fresh produce and irregularity of food supply have been noted in the literature as pantry client barriers to utilizing fresh produce [48]. The current study found consistencies with all three-stakeholder groups reporting barriers in using and preparing unusual produce. Interventions targeting weekly cooking classes within a six-week format have been shown to improve diet quality and decrease food cost within the pantry setting [21] by teaching food preparation skills. Little is known about using a similar program structure targeting PA, and further a holistic program targeting both, PA and nutrition as a comprehensive chronic disease health intervention program.

Staff report more “hyper specificity” in the types of foods clients are requesting due to doctor recommendations through food prescriptions, yet neither staff nor volunteers have the expertise to address these client needs. Thus, trained health educators and/or health coaches could help fill this void [49]. Health professionals could provide services such as pantry shopping assistance, food item identification, recipes, and food skills training that match specific client needs [50]. Due to this gap in expertise among current volunteers and staff, health intervention programming within the food pantry setting would require, either a hired staff member, additional recruitment and training of volunteers, and/or a partnership with local health organizations.

Nutrition and PA knowledge gaps across a diverse range of categories were recognized between all three stakeholder groups. This ranged from healthy cooking on a budget to exercising with limitations, giving direction to health content as an educational component to health intervention programming. Clients advocated for a positive, non-judgmental climate, entailing goal setting and accountability components. This is consistent with elements used within health coaching models that are linked to improvements in health lifestyle behaviors [49]. Health coaching can combine traditional health education strategies with motivational interviewing techniques to increase knowledge, skills, individual motivation, autonomy, and self-efficacy, promoting changes in health behaviors [29]. Last, support for health intervention programing was generated by all three stakeholder groups, particularly among the priority population. By using a formative community participatory approach [33] to gain support and develop intervention components, there will be a greater chance for intervention success and adoption by pantry clients during implementation.

### Study Limitations

The current study only included one county-wide Midwest food pantry with a small sample of the key stakeholders creating generalization limitations. Additionally, the tools included in the study were designed by an investigator and were not first tested for reliability or validity.

## 5. Conclusions

High rates of chronic disease combined with low nutrition and PA literacy among pantry clients demonstrates the need to address health behaviors. In this study, each stakeholder group provided program component recommendations and indicated support for a health intervention program within the food pantry setting. Further research piloting the design and implementation of such a program in the pantry setting is needed. More specifically, design and implementation of a more holistic approach incorporating both nutrition and PA aimed at individual needs and disease prevention. The results will be used to prepare phase two, design and implement a health intervention program within a county-wide Midwest food pantry. Furthermore, key highlights from this research work that could be transferable into the field include:High rates of disease combined with low nutrition and PA literacy highlight the importance of holistic health intervention programming targeting health behaviors and chronic disease among food pantry clients. This includes considering intervention designs that go beyond addressing a single disease (e.g., diabetes) and work within a broader framework to address disease prevention and management (e.g., health coaching).A lack of expertise among volunteers and staff suggests program implementation will require hired staff members, specialized volunteers, and/or partnerships with local health organizations. This warrants the need to build community partnerships and create opportunities for additional training within pantry staff and volunteers to include an ecological approach to intervention design and implementation.Key characteristics of health intervention programming included accountability, incentives and individual attention. Mixed results regarding the program delivery platform lend to hybrid format options (in-person, virtual, group, and individual). Health coaching incorporates elements such as individual attention, social support, motivational interviewing, and accountability that match these intervention characteristics. This approach has been minimally tested in the food pantry setting.All three stakeholder groups recognized individual-level client needs and gaps in programming, aimed at prevention, prior to disease onset. Intervention programming that is focused on individual level need, such as health coaching, can lend to an intervention, which meets both disease management and disease prevention needs of food insecure pantry clients.

## Figures and Tables

**Table 1 nutrients-13-01584-t001:** Pantry client demographic Characteristics and health status.

Demographic Category	Food Pantry Client Characteristics (*n* = 30)	*n*	%
Gender	Male	8	26.7
	Female	22	73.3
Age, years	20–30	4	13.3
	31–40	2	6.7
	41–50	9	30.0
	51–60	9	30.0
	61–73	6	20.0
Race	Caucasian/White	24	80.0
	African American/Black	5	16.7
	Hispanic/Latino	1	3.3
Annual Household Income	<$10,000	13	43.3
	$10,000–$24,999	15	50.0
	$25,000–$49,999	2	6.7
Occupational Status	Working full-time	5	16.7
	Working part-time	4	13.3
	Unemployed, currently seeking	8	26.7
	Unemployed, not currently seeking	7	23.3
	Retired	6	20.0
Education Level	Some high school	1	3.3
	High school graduate or GED	5	16.7
	Some college	14	46.7
	Associate degree	6	20.0
	Bachelor’s degree	3	10.0
	Master’s degree	1	3.3
Number of health conditions (individual)	Zero chronic disease listed	1	3.3
	One chronic disease listed	18	60.0
	More than one chronic disease listed	11	36.7
Specification of health condition (individual)	Diabetes	4	13.3
	High Blood Pressure	11	36.7
	High Cholesterol	7	23.3
	Heart Disease	3	10.0
	Metabolic Syndrome	5	16.7
Overall Health	Excellent	1	3.3
	Very Good	9	30.0
	Good	11	36.7
	Fair	9	30.0
Last Doctor Visit	1–3 months	21	70.0
	4–6 months	5	16.7
	>1 yr.	4	13.3
Health Insurance Status	Insured	22	73.3
	Uninsured	8	26.7
Number of health conditions (household)	Zero chronic disease listed	12	40.0
	One chronic disease listed	13	43.3
	More than one chronic disease listed	5	16.7
Specification of health condition (household)	Diabetes	5	16.7
	High Blood Pressure	7	23.3
	High Cholesterol	5	16.7
	Heart Disease	4	13.3
	Metabolic Syndrome	4	13.3

**Table 2 nutrients-13-01584-t002:** Final convergent Major Themes and Quotes.

Major Theme	Participant Quotes
Nutrition and physical activity barriers	“Mostly the prices [referring to barriers]. The cheaper it is, the less healthy it is. I have walked through a few organic isles, but it is just off the charts, even for food stamps.” (C.1) “I am a single person and I can’t buy a whole chicken, I just want one or two pieces, if I buy a roast, I want to buy a small one. I don’t want to have a whole bunch of spoiled stuff.” (C.9) “The short life of the produce, some of the stuff from the farmers market here will last a week, but some of the stuff from the supermarkets is old.” (C.21) “The hip, back pain or issues [barrier to PA]. I found out I have first stage emphysema so breathing issues.” (C.14) “Honestly, when my depression gets bad I have issues with that [motivation for physical activity] (C.15) “I don’t think people quite understand the importance of nutrition and physical activity because they are just trying to survive.” (V.8) “Sometimes I see the specific recipes and I think there is no way they are going to have those ingredients.” (V.5) “There is a high proportion of individuals who also have physical limitations or physical barriers to physical activity” (S.5) “I have experience direct interactions with clients that view physical activity as perhaps a luxury that they can’t afford yet.” (S.3) “I would say in my experience I get very little interaction with clients who are on a preventative track [related to disease, diet, exercise].” (S.3)
Nutrition and Physical Activity Literacy	Sometimes the knowledge of what to do with certain food. Not knowing how to cook it or what to do with it. Knowing how to use different ingredients or spices.” (C. 26) “A lot of health issues, when you have diabetes or that other stuff, how to incorporate that into your daily life or eating. Foods that you’re able to make to help you with your health challenges like diabetes and other things.” (C.10) “I think I would be interested in learning what type of activities I could do, due to the fact of arthritis in the knees.” (C.20) “It would always be nice to know types of exercises you could do. I was in wrestling in high school, and all we did was like weights and stretch. So that is all I know.” (C.1) “Even stuff like the plate [referencing MyPlate]. It is basic, but it is useful for people to know” (V.5) “Maybe explaining you can walk and it’s still exercise. The little things that are PA and the benefits so like losing weight and the actual health benefits to your heart” (V.3) “I get a lot of recipe questions, or what does this go with, or does this go together, what can I do with these three things.” (S.2) “I think partially, it is a lack of knowledge about preparation, but also there is an assumption that I can’t prepare something a certain way because I don’t have the specialized machinery for it.” (S.3)
Health Status and Lifestyle	“I have PTSD and Depression, high blood pressure, the knee injury, the hand injury, my boyfriend he has high cholesterol. He also has PTSD from Afghanistan and Iraq.” (C.6) “Hip problems and osteoporosis in the hip, stage one emphysema, I have problems that I feel are manageable as far as, yeah, I have been diagnosed with depression and I have been for years.” (C.14) “I have a 13-year-old now that has prediabetes and I am scared to death. We all have ADHD, we control it with medication. I also have schizoaffective disorder. I am living in recovery from drugs and alcohol. My sons are high anxiety. I just began taking antianxiety meds. Because of the stress I was being put under, I am a survivor of child sex abuse, so I am still very affected. It’s more like coming back to the surface. It took just 10 pounds more to make me just start getting really sensitive again.” (C.19) I have had gastric sleeve surgery, so pretty much 80% of everything I eat needs to be protein. Since the surgery, my stomach is real finicky so even stuff I am supposed to eat I can’t. I am limited by what surgery did to me.” (C.2) “I have a thyroid condition that I am supposed to be working on. And I supposed to be on a diet for it. But it’s hard to get the meal planning to get it situated.” (C.4) “People asking about health related things. I am finding more and more hyper specificity for [clients requesting] this is my diet, I have talked to my doctor and they say I need to be eating these specific items, can you help me find those, or do you have any of those?” (S.3) “I keep going back to mental and emotional health. So we know that majority of our clients live in really high stress situations for a number of reasons, so I think food can be, I have seen clients in the past that had food addiction, and this place can be incredibly triggering.” (S.1)
Current Pantry operations and adjustments	“Identification of vegetables that are a bit different. We have had them here before, but they are incredibly passive, so that is people to sign up for SNAP.” (V.9) “Maybe you could be [pantry name] certified to help with that kind of thing [specialty diet recommendations]. Then if somebody has that issue then put it on their profile and you know what volunteers are certified to help with that. So specific volunteers can be certified in certain things and not everything.” (V.3) “If you train more in blood pressure and things like that, it would be nice, but I don’t know if I have time for something like that.” (V.9) “From a programmatic standpoint, I am asked about every type of chronic disease and nutrition for those specific diseases.” (S.5) “It would be kind of cool to have point guides [referencing how to get the most out of your allowable pantry points].” (S.6) “Nutrition facts placed around the place. Like what is the average suggested caloric intake for a day, and sodium intake.” (S.1) “I would love to know more nutritional information [reference to additional training]. Honestly, I don’t know much of any. Maybe, I know my ways of stretching meals, but that doesn’t mean it will work for everybody, like how can I help support.” (S.1)
Health Intervention Program logistics	“I don’t do technology, so I would want face-to-face. I think maybe a lot of people don’t have the money, if they are coming here, they maybe can’t afford internet” (C.3) “You might get more people there if you reward them somehow. Like food or gift certificates to some place. Preferably some place healthy.” (C.3) “Individual [intervention delivery] because everyone has individual difference. Because you have your physical limits, but you also have your health. Some are disabilities. everybody is different.” (C.8) “If you can make it available online face-to-face like through zoom, wouldn’t that be great. Just the touching base.” (C.19) “If you go at them, I guess too forcefully or judgmental, you push them away. Approaching something with a positive outline.” (C.12) “Accountability, you need to be able to hold people accountable. If you don’t do that you aren’t going to get a good result.” (C. 27) “Ideally how to eat healthier. Healthy eating. Do that, but not do it in a patronizing way. It is too often it’s blamed on the individual.” (V.9) “I think that is one of the biggest one. Educating on how to effectively use their points. Some people only have 10 points and they get 4 sandwiches and that is going to last you max of 2 days.” (V.2) “Like the meal kits stuff they come up with [referencing premade meal kits available to clients]. Like super simple stuff that doesn’t require a lot of time or something I can maybe order a lot that [food items] we could use every week.” (S.4) “Budget stuff, I think talking to them about process of food acquisition, do you come here first or do you go to the grocery store first, you should come here first see what you can get with your points, and then build recipes off what you can make with what you get kind of thing.” (S.6) “We have had a lot of people recently asking specifically for items. So they want to do it, but sometimes they can’t. So maybe it would help to get together with the coach and say hey this week we are going to have this, or I can order this, then we can really encourage people to go down this route.” (S.4)

Note: C = Client quote, S = Staff quote, V = Volunteer quote.

## Data Availability

The data presented in this study are available on request from the corresponding author.

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
