# Peer review of "Hunger and Health: Taking a Formative Approach to Build a Health Intervention Focused on Nutrition and Physical Activity Needs as Perceived by Stakeholders"

_nutrients, 2021, doi:10.3390/nu13051584_

Round 1
Reviewer 1 Report
The authors surveyed and interviewed those in a state of food security to coach their entire health through the food pantry.
It notably raised the need for changes in health habits in chronic patients with low health literacy. It is a fascinating study in that various forms of health care programs can fill the blind spots of health and contribute meaningfully to promoting public health through close-up research related to food, a key source of health, beyond traditional medical categories.
If there is information about whether a single family member or a physical disability, it will help build a holistic platform in the future.
Author Response
(x) English language and style are fine/minor spell check required
An additional spelling and grammar check was conducted.
The authors surveyed and interviewed those in a state of food security to coach their entire health through the food pantry. It notably raised the need for changes in health habits in chronic patients with low health literacy. It is a fascinating study in that various forms of health care programs can fill the blind spots of health and contribute meaningfully to promoting public health through close-up research related to food, a key source of health, beyond traditional medical categories. If there is information about whether a single family member or a physical disability, it will help build a holistic platform in the future.
I don't see any additional corrections in this section. Please let me know if I missed something.
Reviewer 2 Report
The manuscript entitled “Hunger and Health: Taking a Formative Approach to Building a Health Intervention Focused on Nutrition and Physical Activity Needs as Perceived by Stakeholders” addresses one main point: understand nutrition and physical activity needs as perceived by food pantry stakeholders (pantry clients, volunteers, and staff) to inform a health intervention approach at a county-wide Midwest food pantry. The topic of research is interesting. However, the authors need to explain why their findings are interesting/important. There is no real evidence provided or examples of practical application which substantiate their claims. Without this, the impact of the results is not going to be realized.
Moreover, there are some points, which I detail bellow, that need to be addressed before publication can be considered:
- A clear definition of food insecurity should be reported.
- Pag 2. Lines 70-72. There is more literature regarding the association between food insecurity and physical activity that could be addressed.
- Results from previous research of the team should be addressed in the introduction.
- What is the novelty of this study?
- how can these results be applied in practical terms?
- how these results can provide different guidelines for creating intervention studies based on nutrition and physical activity, different from those that already exist in the literature
Author Response
- A clear definition of food insecurity should be reported.
I have added this into the introduction, P. 1, lines 28-29
- Pag 2. Lines 70-72. There is more literature regarding the association between food insecurity and physical activity that could be addressed.
I still wasn't able to find much addressing the adult population. I found some additional articles related to prevalence among kids, perceptions, readiness to engage, and physical limitations and disease creating barriers. Additional references provide, P.2, Lines 69- 86.
- Results from previous research of the team should be addressed in the introduction.
Considering the research previously conducted wasn't published, this statement has been removed from the introduction P. 3, line 129-130.
- What is the novelty of this study?
Novelty has been added in the introduction P. 3, lines 129-123. with the statement " This study aims to fill a gap in the literature by 1) understanding more about physical activity behaviors and needs among food insecure adult pantry users; and 2) exploring program components of a comprehensive health intervention that incorporates both physical activity and nutrition as perceived by pantry stakeholders."
- how can these results be applied in practical terms?
This is addressed in the conclusion p. 15, lines 475-487.
- how these results can provide different guidelines for creating intervention studies based on nutrition and physical activity, different from those that already exist in the literature
This is addressed in the conclusion p. 15, lines 475-487. I have expanded on each bullet to further address the novelty.
. Further, key highlights from this research work that could be transferable into the field include:
- High rates of disease combined with low nutrition and PA literacy highlight the importance of holistic health intervention programming targeting health behaviors and chronic disease among food pantry clients. This includes considering intervention designs that go beyond addressing a single disease (e.g. diabetes) and work within a broader framework to address disease prevention and management (e.g. health coaching).
- Lack of expertise among volunteers and staff suggests program implementation will require hired staff members, specialized volunteers, and/or partnerships with local health organizations. This warrants the need for building community partnerships and creating opportunities for additional training with pantry staff and volunteers to include an ecological approach to intervention design and implementation.
- Key characteristics of health intervention programming included accountability, incentives, and individual attention. Mixed results regarding the program delivery platform lend to hybrid format options (in-person, virtual, group, and individual). Health coaching incorporates elements such as individual attention, social support, motivational interviewing, and accountability, that match these intervention characteristics. This approach has been minimally tested in the food pantry setting.
- All three stakeholder groups recognized individual level client need and gaps in programming aimed at prevention prior to disease onset. Intervention programming focused on individual level need, such as health coaching, can lend to an intervention that meets both disease management and disease prevention needs of food insecure pantry clients.
Round 2
Reviewer 2 Report
The authors have done a good job responding to comments on the first version of the manuscript and I thank them for the additions.
Author Response
The reviewer doesn't appear to have additional edits. We appreciate the feedback and improvement suggestions.